# Data Augmentation MCMC for Bayesian Inference from Privatized Data

**Nianqiao Phyllis Ju**
Department of Statistics
Purdue University
West Lafayette, IN 47907
nianqiao@purdue.edu

**Jordan A. Awan**
Department of Statistics
Purdue University
West Lafayette, IN 47907
jawan@purdue.edu

**Ruobin Gong**
Department of Statistics
Rutgers University
Piscataway, NJ 08854
ruobin.gong@rutgers.edu

**Vinayak A. Rao**
Department of Statistics
Purdue University
West Lafayette, IN 47907
varao@purdue.edu

## Abstract

Differentially private mechanisms protect privacy by introducing additional randomness into the data. Restricting access to only the privatized data makes it challenging to perform valid statistical inference on parameters underlying the confidential data. Specifically, the likelihood function of the privatized data requires integrating over the large space of confidential databases and is typically intractable. For Bayesian analysis, this results in a posterior distribution that is doubly intractable, rendering traditional MCMC techniques inapplicable. We propose an MCMC framework to perform Bayesian inference from the privatized data, which is applicable to a wide range of statistical models and privacy mechanisms. Our MCMC algorithm augments the model parameters with the unobserved confidential data, and alternately updates each one. For the potentially challenging step of updating the confidential data, we propose a generic approach that exploits the privacy guarantee of the mechanism to ensure efficiency. We give results on the computational complexity, acceptance rate, and mixing properties of our MCMC. We illustrate the efficacy and applicability of our methods on a naïve-Bayes log-linear model and on a linear regression model.

## 1 Introduction

**Motivation.** Differential privacy [Dwork et al., 2006] presents a formal mathematical framework to protect the confidentiality of individuals and businesses in aggregate data products. It is the state-of-the-art standard for statistical disclosure limitation (SDL), and has become widely adopted by curators of large-scale scientific, commercial, and official databases. Differentially private data products are produced by probabilistic mechanisms that carry proven privacy guarantees. Generally speaking, these mechanisms work by introducing carefully designed random noise into the query of interest, which is an otherwise deterministic function of the underlying database.

The privatization of data products through noise infusion poses a challenge to statistical analysis in the downstream. Statistical estimators are typically complex functions of the data. If instead of the confidential data, the analyst only has access to a probabilistically processed version of them, how can they maintain the statistical validity of the resulting inference?

36th Conference on Neural Information Processing Systems (NeurIPS 2022).

A crucial statistical advantage of differentially private mechanisms over traditional SDL counterparts, such as swapping [Dalenius, 1977], is that their probabilistic design is publicly known. This knowledge allows the data analyst to, at least in theory, accurately account for the privatization mechanism and conduct reliable uncertainty quantification. Nevertheless, it remains a substantial computational challenge to incorporate the privacy procedure into the statistical analysis. The challenge is a wide-spread and varied one, as the extra layer of privacy protection calls for the revision of a wide range of existing statistical methodologies that previously operate on the original, non-privatized data, most of which are neither low-dimensional nor simply structured. This is the challenge we address in this work, in which we develop a general computational framework for practitioners to obtain valid statistical inference based on privatized data.

**Related literature.** Current inferential strategies for privatized data fall into two broad categories. One invokes traditional statistical asymptotics to approximate the sampling distribution of a differentially private statistic, on the grounds that the privacy noise is often asymptotically negligible compared to errors due to sampling [e.g. Smith, 2011, Cai et al., 2021]. These approximations are often inaccurate for finite sample sizes [Wang et al., 2018] and call for specific handling to incorporate the privacy mechanism [e.g. Gaboardi et al., 2016, Wang et al., 2015, Gaboardi and Rogers, 2018].

The second category recognizes (as Section 2 will explain) that the marginal likelihood of the model parameters in (3) is central to the problem of inference from privatized data. The marginal likelihood/marginal posterior distribution requires a potentially high-dimensional integral over the space of unobserved confidential databases, and one that is analytically tractable only in a few, simple settings [e.g., Awan and Slavković, 2018, 2020]. Typically, one must resort to either approximating it or sampling from it using Monte Carlo methods. Markov chain Monte Carlo (MCMC) techniques have been proposed for specific privacy mechanisms and data generating models. Karwa et al. [2017] propose an MCMC procedure for inference on exponential random graph models, Bernstein and Sheldon [2018, 2019] devise MCMC methods designed to handle the low-dimensional latent sufficient statistics from exponential family models and linear regression, and Schein et al. [2019] design an MCMC algorithm for the Poisson factorization model when data are locally privatized under the double geometric mechanism. Gong [2022] shows that for certain differentially private statistics, approximate Bayesian computation (ABC) can give samples that are exact with respect to the marginal likelihood and the Bayesian posterior. In addition, when the statistical model for the confidential data is fully parametric, the parametric bootstrap may be used to produce inference accompanied by uncertainty quantification with better accuracy than asymptotic approximation [e.g. Gaboardi et al., 2016, Ferrando et al., 2022]. Variational Bayesian analysis [Karwa et al., 2016] is another alternative which invokes a non-asymptotic approximation to the posterior distribution. These solutions are either approximations or hold only for specific settings.

**Our contribution.** We develop a general-purpose MCMC framework to perform Bayesian inference on the model parameters underlying the privatized data. Our framework allows us to overcome the intractable marginal likelihood resulting from privatization, and is applicable to a wide range of statistical models and privacy mechanisms. The resulting MCMC algorithms are *exact*, in that they target the posterior distribution precisely, without involving any approximation.

Our approach allows data analysts to leverage existing inferential tools designed for non-private data. It can be viewed as a flexible, user-friendly wrapper that migrates existing MCMC algorithms for non-private data to the setting of privatized data access, requiring no further algorithm design or tuning. The sampler, formally a Metropolis-within-Gibbs sampler, is presented in Algorithm 1. It is general-purpose, requiring only that the analyst can 1) sample from the statistical model for the confidential data and 2) can evaluate the probability density of the noise induced by the privacy mechanism. The algorithm augments the model parameters with the unobserved confidential data, and alternately updates each one conditioned on the other. While the imputation of an entire unobserved database might appear daunting, we demonstrate how knowledge of the privacy mechanism can be exploited to confer performance guarantees to the proposed MCMC algorithm. We provide theoretical results for the computational complexity, Metropolis-Hastings acceptance rate, and mixing properties. In particular, the higher the privacy, the more rapid is our algorithm's exploration of the parameter space. We also identify a common structure present in many popular privacy mechanisms, which we term *record additivity*. The record additivity of a privacy mechanism ensures that each round of our sampler has the same order of run-time as that of many non-private samplers. We illustrate the efficacy and applicability of our methods on a privatized naïve-Bayes log-linear model and a linear regression model with clamped and privatized input.

## 2 Problem Setup

Let $x = (x_1, \ldots, x_n) \in \mathbb{X}^n$ denote the confidential database, containing $n$ records. We assume these records are independent and identically distributed (i.i.d.) draws from a statistical model $f(\cdot \mid \theta)$, though this can be relaxed. The goal of the analyst is to conduct statistical inference on the unknown model parameter $\theta \in \Theta$. A Bayesian analyst represents *a priori* beliefs about $\theta$ with a prior probability distribution $p(\theta)$, and seeks to compute a posterior distribution $p(\theta \mid x) \propto p(\theta) f(x \mid \theta)$ that updates their beliefs in light of the observations $x$. In many modern applications, this posterior distribution is intractable, and it is common for analysts to represent it using samples drawn via some MCMC algorithm. In this work, we will assume access to such a posterior sampling method:

**Assumption 1.** *The analyst has available a Markov kernel that targets $p(\theta \mid x) \propto p(\theta) f(x \mid \theta)$, the posterior distribution over the model parameters given the confidential database $x$.*

**Differential privacy.** Our work here focuses on the following departure from the usual Bayesian setting: instead of observing the database $x$, we observe a privatized data product or query, denoted as $s_{\mathrm{dp}}$. The quantity $s_{dp}$ is typically of much lower dimensionality that the database $x$, and is probabilistically generated based on data $x$ through a *privacy mechanism*, written as $\eta(\cdot \mid x)$. The privacy mechanism $\eta$ is said to be $\epsilon$-*differentially private* ($\epsilon$-DP) [Dwork et al., 2006] if for all values of $s_{\mathrm{dp}}$, and for all 'neighboring' databases $(x, x') \in \mathbb{X}^n \times \mathbb{X}^n$ differing by one record (denoted by $d(x, x') \leq 1$), the probability ratio is bounded:

$$\frac{\eta(s_{\mathrm{dp}} \mid x)}{\eta(s_{\mathrm{dp}} \mid x')} \leq \exp(\epsilon), \quad \epsilon > 0. \tag{1}$$

The parameter $\epsilon$ is called the *privacy loss budget*, and controls how informative $s_{\mathrm{dp}}$ is about $x$. Large values of $\epsilon$ guarantee less privacy, while $\epsilon = 0$ corresponds to perfect privacy. A simple and widely used $\epsilon$-differentially private mechanism is the *Laplace mechanism*: for a deterministic query $s : \mathbb{X}^n \to \mathbb{R}^m$, the privatized query is defined as $s_{\mathrm{dp}} = s(x) + u$, where $u = (u_1, \ldots, u_m)$ are i.i.d. Laplace variables. The scale parameter of the Laplace distribution is inversely proportional to $\epsilon$ (more privacy requires more noise), and directly proportional to $\Delta(s) = \max_{(x, x') \in \mathbb{X}^n \times \mathbb{X}^n; d(x, x') \leq 1} \|s(x) - s(x')\|_1$, the $\ell_1$ *(global) sensitivity* of $s$ (the more sensitive the confidential query is to changes in one record of the database, the more noise we need).

Our methodology requires that the privacy mechanism $\eta$ is known and can be evaluated. This is true of $\epsilon$- (or *pure*) DP, as well as common variants such as $(\epsilon, \delta)$- (or *approximate*) DP, *zero-concentrated* DP (zCDP) [Dwork and Rothblum, 2016, Bun and Steinke, 2016], and *Gaussian*-DP [Dong et al., 2021]. To ensure computational efficiency, we make the following additional assumption.

**Assumption 2** (Record Additivity). *The privacy mechanism can be written in the form $\eta(s_{\mathrm{dp}} \mid x) = g(s_{\mathrm{dp}}, \sum_{i=1}^{n} t_i(x_i, s_{\mathrm{dp}}))$ for some known and tractable functions $g, t_1, \ldots, t_n$.*

We refer to privacy mechanisms that satisfy Assumption 2 as *record-additive*. An implication of record additivity is that after changing one record in $x$, we do not have to scan the entire database to reevaluate $\eta$. This is satisfied by many commonly used mechanisms, two important examples being: 1) mechanisms that add data-independent noise to a query of the form $s = \sum_{i=1}^{n} s_i(x_i)$, such as the sample mean, sample variance-covariance, and sufficient statistics of an exponential family distribution (see Sections 4 and 5 for examples), and 2) mechanisms designed to optimize empirical risk functions of the form $u(x, s_{dp}) = \sum_{i=1}^{n} u_i(x_i, s_{dp})$, such as the exponential mechanism [McSherry and Talwar, 2007], $K$-norm gradient mechanism [Reimherr and Awan, 2019], objective perturbation [Chaudhuri et al., 2011, Kifer et al., 2012], and functional mechanism [Zhang et al., 2012].

**Doubly intractable Bayesian inference from privatized data.** Without access to the confidential database $x$, and given only the privatized query $s_{\mathrm{dp}}$, the Bayesian analyst is now concerned with the following posterior distribution:

$$p(\theta \mid s_{\mathrm{dp}}) \propto p(\theta) p(s_{\mathrm{dp}} \mid \theta). \tag{2}$$

Here, $p(s_{\mathrm{dp}} \mid \theta)$ is the *marginal likelihood* of $\theta$, integrating over all possible confidential databases:

$$p(s_{\mathrm{dp}} \mid \theta) = \int_{\mathbb{X}^n} \eta(s_{\mathrm{dp}} \mid x) f(x \mid \theta) \, dx. \tag{3}$$

The marginal likelihood contributes all the information that is available in the privatized observation $s_{\mathrm{dp}}$ about the parameter $\theta$, and is the foundation to statistical inference using privatized statistics

[Williams and McSherry, 2010]. The posterior distribution (2) reveals that the inferential uncertainty about the parameter $\theta$ consists of three contributing sources: 1) prior uncertainty as encoded in $p(\theta)$, 2) sampling (or modeling) uncertainty of the confidential database as reflected in $f$, and 3) uncertainty due to privacy as induced by the probabilistic mechanism $\eta$.

We now come to the core challenge to address in this work: the marginal likelihood in (3) calls for an integral over the entire space of possible input databases $x \in \mathbb{X}^n$. This is usually computationally challenging, especially if the privacy mechanism is not a function of a low-dimensional sufficient statistic. If the integral underlying the marginal likelihood is intractable, then $p(s_{\mathrm{dp}} \mid \theta)$ cannot be analytically evaluated. This makes the corresponding posterior distribution $p(\theta \mid s_{\mathrm{dp}})$ of (2) *doubly* intractable [Murray et al., 2012] in the sense that it cannot be analytically evaluated even up to a normalizing constant. Thus, traditional MCMC techniques are inapplicable and inference strategies devised for privatized statistics must tame this possibly high-dimensional integration problem.

## 3 Data Augmentation MCMC for Inference from Privatized Data

In this paper, we present a simple, efficient, and general *data augmentation* MCMC [Tanner and Wong, 1987, Van Dyk and Meng, 2001] framework, allowing practitioners to perform valid Bayesian inference on a wide-range of data models and privacy mechanisms. Our approach is to augment the MCMC state space with the latent confidential database $x$, so that the stationary distribution is the *joint* posterior distribution

$$p(\theta, x \mid s_{\mathrm{dp}}) \propto p(\theta)f(x \mid \theta)\eta(s_{\mathrm{dp}} \mid x). \tag{4}$$

Marginally, the $\theta$ samples produced by such an algorithm follow the posterior $p(\theta \mid s_{\mathrm{dp}})$ in (2). Our sampler is *exact*, targeting the marginal posterior distribution $p(\theta|s_{\mathrm{dp}})$ without any approximation error, despite the fact that the marginal likelihood (3) is intractable.

Our approach of imputing the latent confidential database $x$ is motivated by two factors: 1) we wish our algorithm to be *general-purpose*, applicable to a wide range of models and privacy mechanisms, and 2) we wish our algorithm to inherit guarantees on mixing performance from guarantees of the privacy mechanism. Towards these ends, we do not assume any specific form of the underlying model of $x$ and the privacy mechanism beyond Assumptions 1 and 2 respectively. In this light, our contribution can be viewed as a flexible wrapper that allows existing MCMC algorithms for models of the confidential data to be extended to settings where the data is now protected by some privacy mechanism. Though imputing the confidential dataset might appear to present a significant challenge, we show that properties of the mechanism can be exploited to give performance guarantees on our sampling scheme, and show that it has a runtime of the same order as the non-private sampler.

In what follows, we outline our proposed MCMC algorithm, derive guarantees on the runtime and acceptance rate of the algorithm, and provide mild conditions for the proposed samplers to be ergodic, as well as additional conditions for our sampler to achieve geometric rates of convergence.

### 3.1 A Privacy-Aware Metropolis-within-Gibbs Sampler

Our approach to sample from the joint posterior distribution $p(\theta, x \mid s_{\mathrm{dp}})$ is through a sequence of alternating Gibbs updates. Let $(x^{(t)}, \theta^{(t)})$ denote the state of the Gibbs sampler at the $t$-th iterations. Each iteration of the Gibbs sampler entails two steps:
(**Step 1**) sample $\theta^{(t+1)}$ from $p(\cdot \mid x^{(t)}, s_{\mathrm{dp}})$, and (**Step 2**) sample $x^{(t+1)}$ from $p(\cdot \mid \theta^{(t+1)}, s_{\mathrm{dp}})$.

The conditional distribution in Step 1 simplifies as $p(\theta|x^{(t)}, s_{\mathrm{dp}}) = p(\theta|x^{(t)})$, highlighting why data-augmentation is useful: this conditional distribution is independent of the privacy mechanism, and we can use existing sampling algorithms (Assumption 1) for the confidential data. We note that with the exception of a few models, such as simple models with conjugate priors, it is usually not possible to directly sample from $p(\theta|x)$. Assumption 1 however only requires that we can *conditionally* simulate a new value of $\theta$ from a Markov kernel that has $p(\theta|x^{(t)})$ as its stationary distribution. Our overall Gibbs sampler then becomes a Metropolis-within-Gibbs sampler [Gilks et al., 1995], that nevertheless targets the joint posterior $p(\theta, x|s_{\mathrm{dp}})$.

Step 2 is the data-augmentation step, and connects the statistical model and the privacy mechanism on $x$. Again, we cannot expect to produce a conditionally independent sample of the latent database from

$p(x \mid \theta, s_{\mathrm{dp}})$, as this is model and mechanism dependent. Instead, we take the much more tractable approach of cycling through the elements of latent database $x$, sequentially updating $x$ one element of a time. Writing $x_{-i} = (x_1, \ldots, x_{i-1}, x_{i+1}, \ldots, x_n)$ to denote the vector $x$ excluding the $i$th element, step 2 then consists of the sequence of updates $p(x_1 \mid \theta, x_{-1}, s_{\mathrm{dp}}), p(x_2 \mid \theta, x_{-2}, s_{\mathrm{dp}}), \ldots, p(x_n \mid \theta, x_{-n}, s_{\mathrm{dp}})$. The complete sweep can be viewed as a dependent update of the latent database $x$ that targets the conditional distribution $p(x \mid \theta, s_{\mathrm{dp}})$.

Before we specify our complete sampler in Algorithm 1, we first address the following questions: (Q1) *What is the performance loss from updating $x$ one element at a time, rather than jointly?*, and (Q2) *How can we efficiently carry out the conditional updates $p(x_i \mid \theta, x_{-i}, s_{\mathrm{dp}})$, $i = 1, \ldots, n$?*

Q1 concerns whether the dependence of $x_i$ given $(x_{-i}, s_{\mathrm{dp}})$ is so strong as to impede efficient exploration of the $\mathbb{X}^n$-space and cause poor mixing. Here we note that the privacy mechanism limits the change in the likelihood $\eta(s_{\mathrm{dp}} \mid x)$ when one element of $x$ is changed, and therefore limits the coupling between $x_i$ and $x_{-i}$. This suggests a Gibbs sweep through the latent database $x$ will not suffer from poor mixing.

---

**Algorithm 1** One iteration of the privacy-aware Metropolis-within-Gibbs sampler

1. Conditional update of $p(\theta \mid x)$ using the kernel from Assumption 1.
2. For each $i = 1, 2, \ldots$, sequentially update $x_i \mid x_{-i}, \theta, s_{\mathrm{dp}}$.
   (a) Propose $x_i^\star \sim f(\cdot \mid \theta)$.
   (b) Update $t(x^\star, s_{dp}) = t(x, s_{dp}) - t_i(x_i, s_{dp}) + t_i(x_i^\star, s_{dp})$ according to Assumption 2.
   (c) Accept the proposed state with probability $\alpha(x_i^\star \mid x_i, x_{-i}, \theta)$ given by:

$$\alpha(x_i^\star \mid x_i, x_{-i}, \theta) = \min\left\{ \frac{\eta(s_{dp} \mid x_i^\star, x_{-i})}{\eta(s_{dp} \mid x_i, x_{-i})}, 1 \right\} = \min\left\{ \frac{g(s_{\mathrm{dp}}, t(x^\star, s_{\mathrm{dp}}))}{g(s_{\mathrm{dp}}, t(x, s_{\mathrm{dp}}))}, 1 \right\}. \quad (5)$$

---

Q2 recognizes that the conditionals $p(x_i \mid \theta, s_{\mathrm{dp}}, x_{-i})$ are model- and mechanism-specific, and simulating from these is challenging in most settings. For this, we take the following simple approach in Algorithm 1: at each step, we propose $x_i$ from the model $f(x \mid \theta)$, and accept it with the appropriate Metropolis-Hastings acceptance probability (5). Observe that our choice of proposal distribution is independent of the privacy mechanism, and ignores the privatized data $s_{\mathrm{dp}}$ as well as all other elements $x_{-i}$. Despite being simple and general-purpose, we show in Proposition 3.1 that for $\epsilon-$DP, we can lower-bound the acceptance probability of proposals produced this way by $\exp(-\epsilon)$. This lower bound is key to efficiency: despite the unconstrained nature of the proposal distribution, we can guarantee a minimum acceptance probability. These two facts suggest our sampler will explore the space of databases relatively quickly.

**Proposition 3.1.** *For a pure $\epsilon$-DP privacy mechanism $\eta$, the acceptance probability $\alpha$ from Equation (5) satisfies* $\alpha(x_i^\star \mid x_i, x_{-i}, \theta) \geq \exp(-\epsilon)$, *for all $\theta, x_{-i}, x_i, x_i^*$.*

The privacy loss budget $\epsilon$ is usually understood to be a small constant, which privacy experts recommend be between .01 and 1 [Dwork, 2011]. When $\epsilon = 1$, Proposition 3.1 ensures that the acceptance rate in Algorithm 1 is no less than 36.7%, and as $\epsilon$ approaches zero, the bound on the acceptance rate approaches one. Intuitively, this is because as $\epsilon$ decreases, the distribution of the privatized data $s_{\mathrm{dp}}$ depends less and less on any individual element of the database.

The simplicity of our approach arises through a decoupling of the data model from the privacy mechanism: the former is used to update $\theta$ and propose $x_i$'s, while the latter is used to calculate the acceptance probabilities. The next result formalizes the computational efficiency of our approach. Specifically, for any record-additive mechanism, one iteration of our algorithm requires $O(n)$ operations, where $n$ is the size of the latent database. Essentially, this arises because of Assumption 2, which allows the acceptance probability in (5) to be calculated in $O(1)$ time.

**Proposition 3.2.** *The Gibbs sampler described in Algorithm 1 requires $O(n)$ number of operations to update the full latent database according to $p(x \mid \theta, s_{\mathrm{dp}})$.*

Note that even without privacy, one round of an MCMC procedure typically takes $O(n)$ time. This is because updating $\theta$ given the confidential data requires computing the data likelihood $f(x \mid \theta) = \prod_{i=1}^n f(x_i \mid \theta)$, an $O(n)$ operation in general. Thus, as a result of the mild and typical condition that

$\eta$ is record-additive, our MCMC procedure enjoys the *same order* of runtime as the original MCMC algorithm for confidential data.

The previous two results are key to understanding the efficiency of our approach. In the next section we formally establish geometric ergodicity of the sampler in Theorem 3.3 and 3.4.

**Computational complexity.** The i.i.d. assumption on the records ensures that step 2a of Algorithm 1 take $O(1)$ time, though this assumption can easily be weakened. Assumption 2 allows steps 2b and 2c to also takes $O(1)$ time. Overall, step 2 of our algorithm then takes $O(n)$ (rather than $O(n^2)$) time as stated in Proposition 3.2. This matches the typical per-iteration cost of samplers for the non-private posterior distribution required in step 1. Thus, the overall cost of an iteration of our MCMC sampler is $O(n)$, which is typical when dealing with datasets of size $n$.

## 3.2 Ergodicity of the Privacy-Aware Sampler

Ergodicity ensures the MCMC chain converges to the posterior distribution in total variation distance. [Tierney, 1994], which is essential for an MCMC sampler to consistently estimate functionals of the posterior distribution. In Theorem 3.3, we provide mild and sufficient conditions for our proposed Metropolis-within-Gibbs sampler to be ergodic.

**Theorem 3.3.** *Under conditions A1 - A3 below, the Metropolis-within-Gibbs sampler of Algorithm 1 on the joint space $(\mathbb{X}^n \times \Theta)$ is ergodic and it admits $p(x, \theta \mid s_{\mathrm{dp}})$ as the unique limiting distribution.*

*A1. The prior distribution is proper and $p(\theta) > 0$ for all $\theta$ in $\Theta = \{\theta \mid f_\theta(x) > 0$ for some $x\}$.*
*A2. The model is such that the set $\{x : f(x \mid \theta) > 0\}$ does not depend on $\theta$.*
*A3. The privacy mechanism satisfies $\eta(s_{\mathrm{dp}} \mid x) > 0$ for all $x \in \mathbb{X}^n$.*

We prove this in the supplementary material by verifying invariance, aperiodicity, and irreducibility. Conditions A1-A3 concern model specification, prior specification, and privacy noise. These mild assumptions are typically true and are easy to verify. While there are some mechanisms, such as the release-one-at-random mechanism, which satisfy approximate-DP but which violate A3 [Barber and Duchi, 2014], most privacy mechanisms of interest satisfy A3. It is easy to verify that if $\eta$ satisfies $\epsilon$-DP, zero-concentrated DP, or Gaussian-DP, then property A3 is guaranteed.

Next, we establish conditions for Algorithm 1 to be geometrically ergodic [Rosenthal, 1995, Roberts and Rosenthal, 1998]. A chain is said to be *geometrically ergodic* if its total variation distance to the target has a geometrically decaying upper bound. Geometric ergodicity is a desirable property since it provides a rate on convergence to the stationary distribution, guaranteeing central limit theorems, and allowing for the computation of asymptotically valid standard errors.

For simplicity, we focus on the situation where one can directly sample from the conditional posterior $p(\theta \mid x)$. This is an important and common case, relevant when either $\theta$ is low-dimensional or where one can place conjugate priors on $\theta$. Both applications we present in this work, a log-linear model in Section 4, and a linear regression model in Section 5, fall under this setting.

**Theorem 3.4.** *Assume that in step 1 of Algorithm 1, one can directly sample from $p(\theta \mid x)$. Under A1-A3 of Theorem 3.3, the resulting $(x, \theta)$ chain, as well as the marginal chains, are geometrically ergodic if $\eta$ satisfies $\epsilon$-DP and there exists $0 < a \leq b < \infty$ such that $a \leq f(x \mid \theta) \leq b \quad \forall \theta, x$.*

To prove Theorem 3.4, we verify the drift and minorization conditions for component-wise Gibbs samplers in Theorem 8 of [Johnson et al., 2013]. See the supplementary material for details.

Unsurprisingly, geometric ergodicity requires stronger assumptions than just ergodicity. The first assumption concerning the ability to sample directly from $p(\theta|x)$ can be avoided, but for the sake of clarity we do not try to relax it, since we are mostly concerned with the interface with the privacy mechanism. The second assumption on the boundedness of the likelihood is stronger, but also typical. A common way to achieve bounded likelihoods is to require the sample space $\mathbb{X}$ (and typically also $\Theta$) to be bounded. In many real-world settings, such bounds exist, even if they may be very loose.

# 4 Naïve Bayes Log-Linear Model

Log-linear models are often used to model categorical data, a popular instance being the naïve Bayes classifier. Following Karwa et al. [2016], we consider the following model: $x = (x_1, \ldots, x_K)$

is the input *feature-vector*, with each $x_k$ taking values in $\{1, 2, \ldots, J_k\}$, and $y$ is the output *class* taking values in $\{1, 2, \ldots, I\}$. Each input-output pair $(x, y)$ forms one record in our confidential database, the entire database consisting of $N$ i.i.d. copies of $(x, y)$. The naïve Bayes classifier assumes that $P(x \mid y) = \prod_{k=1}^{K} P(x_k \mid y)$, the model parameters being $p_{ij}^k = P(x_k = j | y = i)$ and $p_i = P(y = i)$. The sufficient statistics of this model are $n_{ij}^k = \#(y = i, x_k = j)$, which count the number of class-feature co-occurrences. This will form our confidential query $s$, which we privatize by adding Laplace noise to each of the $n_{ij}^k$. The resulting quantity $s_{dp}$, consisting of the noisy counts $m_{ij}^k = n_{ij}^k + L_{ijk}$, is what we release. When $L_{ijk} \overset{\text{i.i.d.}}{\sim} \text{Laplace}(0, 2K/\epsilon)$, the output $s_{\text{dp}}$ satisfies $\epsilon$-DP. Placing a $\text{Dirichlet}(2, \ldots, 2)$ on all parameter vectors, our goal is to obtain the marginal posterior distribution of $p, p_{i-}^k \mid s_{dp}$. While Karwa et al. [2016] approximate this private posterior distribution using variational Bayes methods, our MCMC procedure is able to target the exact private posterior distribution.

**Simulation setup.** We perform several simulation experiments where we apply our MCMC samplers to the log-linear model described above. For the simulation, we set $N = 100$ (number of records), $I = 5$ (number of classes), $K = 5$ (number of features), and $J_k = 3$ for all $k = 1, \ldots, K$ (possible values for each feature). We evaluate our sampler for privacy levels corresponding to $\epsilon \in \{.1, .3, 1, 3, 10\}$.

**Posterior mean.** We generate one non-private dataset from the model, and hold it fixed. We then create 100 private queries $s_{\text{dp}}$ at each $\epsilon$ value, and for each $s_{\text{dp}}$ we run Algorithm 1 for 10000 iterations. We discard the first 5000 iterations as burn-in. Finally, for each chain, we calculate the posterior mean. Figure 1a plots the 100 different posterior means for each $\epsilon$-value for the parameters $p_i = P(Y = i)$ for $i = 1, \ldots, 5$. In this plot, the solid horizontal lines indicate the non-private posterior means, and the dashed horizontal lines indicate the prior means for each parameter. We see that as $\epsilon$ approaches zero, the posterior mean approaches the prior mean, reflecting the intuition that we learn less from the data as the privacy budget gets smaller. On the other hand, as $\epsilon$ increases, we see that the private posterior mean approaches the non-private posterior mean, which reflects the fact that as $\epsilon$ grows, we learn approximately the same from the data as if there were no privacy mechanism.

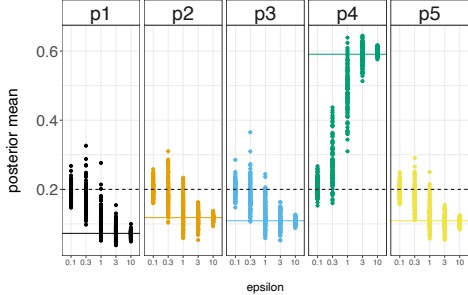
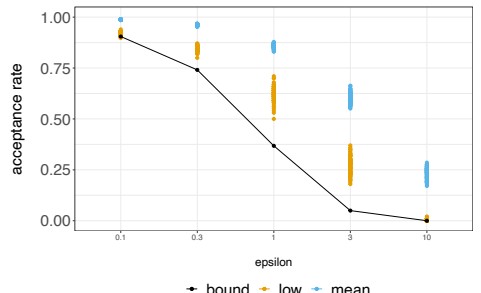

(a) Posterior means for the log-linear model. The solid horizontal lines indicate the non-private posterior means, and the dashed lines at .2 indicate the prior means.

(b) Observed acceptance rates for the log-linear model. The blue (above) point clouds indicate the average acceptance rate, and the orange (below) points indicate the observed minimum acceptance of each chain. The solid black line is the lower bound of Proposition 3.1.

**Acceptance rate.** Using the same simulation setup as for the posterior mean, we calculate the average and minimum acceptance rate of Step 2 in Algorithm 1. Since the privacy mechanism satisfies $\epsilon$-DP, we know that $\exp(-\epsilon)$ is a lower bound on these acceptance rates. In Figure 1b, we confirm this bound, and see that the average acceptance rate is significantly higher than this lower bound. This suggests that the chain mixes even faster than indicated by Proposition 3.1.

**Coverage of credible intervals.** For the next experiment, we sample a set of parameters from the prior, and hold this fixed. Then for each $\epsilon$ value, we produce 100 non-private datasets $(n_{ij}^k)$, one private dataset $(m_{ij}^k)$ for each non-private one, and then run a chain for 10,000 iterations discarding

the first 5000 iterations for burn-in. From each chain, we produce a $90\%$ credible interval for each $p_i = P(Y = i)$, and calculate the empirical coverage which is reported in Table 1.

| $\epsilon$ | $p_1 = .097$ | $p_2 = .148$ | $p_3 = .145$ | $p_4 = .446$ | $p_5 = .163$ |
|---|---|---|---|---|---|
| .1 | 1 | 1 | 1 | **.36** | 1 |
| .3 | .97 | 1 | 1 | **.59** | 1 |
| 1 | .94 | .99 | .97 | **.83** | .98 |
| 3 | .95 | .91 | .97 | .89 | .93 |
| 10 | .92 | .88 | .94 | .92 | .9 |

Table 1: Coverage of $p_i = P(Y = i)$ for the log-linear model at different $\epsilon$. Top row is the true data generating parameter values. Coverage is based on 100 replicates.

At a sample size of only $N = 100$, we do not expect the coverage of the credible intervals to match the nominal level of $90\%$, but we see in Table 1 that most of the coverage values are above .9. Notable exceptions are the coverage of $p_4$ when $\epsilon$ is small. This may be because when $\epsilon$ is small, the private posterior is approximately equal to the prior, which is centered at .2; however $p_4$ is significantly further from .2 than the other parameters, which may explain why the coverage is low in this case.

## 5 Linear Regression

Next, we consider ordinary linear regression with $n$ subjects and $p$ predictors. We write $x_0$ for the matrix of predictors excluding the intercept columns, $x = (\underline{1}, x_0)$ for the matrix including the intercept, and $y$ for the vector of outcomes. We model the explanatory variables $x_0$ as $x_0^i \overset{\text{i.i.d.}}{\sim} \mathcal{N}_p(m, \Sigma)$ for $i = 1, \ldots, n = 100$, with $y|x$ given by $\mathcal{N}_n(x\beta, \sigma^2 I_n)$. Here $I_n$ is the $n \times n$ identity matrix and $\mathcal{N}_n$ denotes the $n$-dimensional multivariate Normal distribution. The parameters of interest are $\beta$, the $(p+1)-$dimension vector of regression coefficients, with $\sigma, m$ and $\Sigma$ assumed known. We use independent $\mathcal{N}(0, 2^2)$ priors for the components of $\beta$.

To achieve $\epsilon$-DP via the Laplace mechanism, we require a finite global sensitivity. To achieve this, standard practice in the DP literature is to bound each predictor and response variable in a data-independent fashion. The bounds chosen by the privacy expert are $[a_i, b_i]$ for each instance of $x_0^i$ and $[a_y, b_y]$ for the entries of $y$, and these values are shared with the analyst.

**Definition 5.1.** For a real value $z$, and $a \leq b$, define the *clamp function* $[z]_a^b := \min\{\max\{z, a\}, b\}$. If $z$ is a vector of length $d$, we use the same notation to apply an entry-wise clamp: $[z]_a^b := ([z_1]_a^b, [z_2]_a^b, \ldots, [z_d]_a^b)^\top$.

Before adding noise for privacy, we first clamp the predictors and response, and then normalize them to take values in $[-1, 1]$: $\widetilde{x}_0^i := (b_i - a_i)^{-1} 2([x_0^i]_{a_i}^{b_i} - a_i) - 1$ and $\widetilde{y} := (b_y - a_y)^{-1} 2([y]_{a_y}^{b_y} - a_y) - 1$. Call $\widetilde{x} := [\underline{1}, \widetilde{x}_0^1, \widetilde{x}_0^2, \ldots, \widetilde{x}_0^p]$ and $s := (\widetilde{x}^\top \widetilde{y}, \widetilde{y}^\top \widetilde{y}, \widetilde{x}^\top \widetilde{x})$. The $s$ is the summary statistic to which we will add noise for privacy. The $\ell_1$ sensitivity of $s$ (ignoring duplicate entries of $\widetilde{x}^\top \widetilde{x}$, and the constant entry $(\widetilde{x}^\top \widetilde{x})_{1,1}$ is $\Delta = p^2 + 3p + 3$. To satisfy $\epsilon$-DP, we add independent $\mathrm{Laplace}(0, \Delta/\epsilon)$ noise to each of the $d = \frac{1}{2}(p+1)(p+2) + (p+1)$ unique entries of $x$, which gives our final private summary $s_{dp}$. We notice that $s$ is an additive function and each individual's contribution to $s$ is $t(x_i, y_i) = ((\widetilde{x}^i)^\top \widetilde{y}_i, \widetilde{y}_i^2, (\widetilde{x}_i)^\top \widetilde{x}_i)$. This mechanism producing $s_{\mathrm{dp}}$ is record-additive.

**Simulation setup.** Our experiments focus on posterior inference about $\beta$ based on $S_{dp}$. For simplicity, we fix other parameters $\sigma^2, m, \Sigma$ at the true data generating parameters (reported in the supplementary materials). When they are unknown, the posterior distributions of these parameters can be estimated by our Gibbs sampler as well. Confidential predictors and responses are clamped with bounds $b = 10$ and $a = -10$. Given a confidential database $(x, y)$, the posterior distribution of $\beta$ is multivariate Normal and can be sampled directly with a runtime linear in $n$.

**Posterior mean.** We generate one confidential dataset $(x, y)$ and hold it fixed. At each $\epsilon$ value, we create 100 private outputs $s_{\mathrm{dp}}$ and run Gibbs samplers for 10,000 iterations targeting the posterior $\beta \mid s_{\mathrm{dp}}$, discarding the first 5000 iterations. We plot the 100 different posterior means of $\beta$ in Figure 2. In this plot, the solid horizontal lines indicate posterior means given confidential data $(x, y)$, which

we do not expect to fully recover due to clamping. The posterior quantities display the same trend with respect to change in privacy level as observed in Figure 1b. The other experiments from Section 4 were also run on this linear regression model, and produced similar results. Simulation details, plots and discussion are in the supplementary materials.

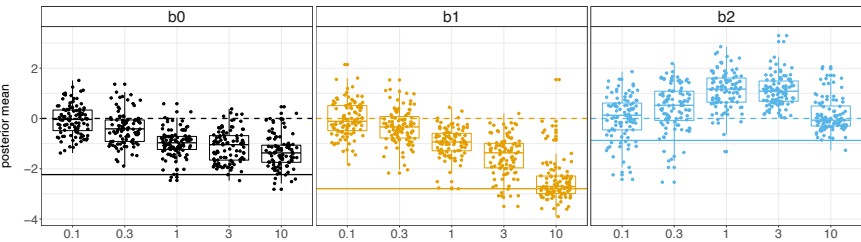

Figure 2: Posterior mean for private linear regression $\beta \mid s_{\mathrm{dp}}$ with fixed confidential data. The solid horizonal lines indicate the confidential data posterior means, and the dashed lines indicate 0.

## 6  Discussion

We proposed a novel, but simple sampling procedure for parameter inference models where one only has access to privatized data. Our approach is an MCMC sampler that targets the posterior distribution $p(\theta \mid s_{\mathrm{dp}})$, and which leverages existing samplers for the non-private posterior $p(\theta \mid x)$, as well as the structure of the privacy mechanism. The result is a simple wrapper for practitioners to obtain valid statistical inference from privatized data using the same models for the unobserved confidential data. As a side product, our algorithm also produces multiple copies of the confidential database from the posterior $p(x \mid s_{\mathrm{dp}})$. These posterior predictive draws could be useful when one is interested in inferring properties of $x$ as well. Although we did not discuss this, our data augmentation scheme can also potentially enable frequentist analysis through the Monte Carlo expectation-maximization algorithm.

We acknowledge some limitations of present work. First, we point out that strong assumptions such as bounded parameter space $\Theta$ and sample space $\mathbb{X}$ are required to establish geometric ergodicity of the Gibbs sampler in Theorem 3.4. They mostly reflect the current state of MCMC mixing results, and can likely be relaxed, at the cost of a more complex theorem statement and proof. Second, our current proposal for updating $x_i \mid x_{-i}, \theta, s_{\mathrm{dp}}$ only tailors to the model $f(\cdot \mid \theta)$ and it is not customized for $s_{\mathrm{dp}}$ yet. In the future, we might be able to design algorithms that also incorporate the privatized output $s_{\mathrm{dp}}$ in these proposals. Third, we point out while our method exploits privacy to control the correlation between the components of $x$, it is still susceptible to correlation between $x$ and $\theta$. This can potentially cause poor mixing in practice, despite geometric convergence rate. While in simple problems, this can be fixed by reparameterization, we plan to develop MCMC algorithms for this setting in follow-up studies. A similar potential issue is that, as a wrapper, our algorithm may be inefficient when the chosen sampler for $p(\theta \mid x)$ does not mix well, especially in some high-dimensional settings. We emphasize though that our method is intended for settings where there currently exists efficient samplers for $p(\theta \mid x)$, and when this is not the case, alternative approaches may be needed.

Finally, while the proposed algorithm converges so long as the privacy mechanism $\eta$ is known, irrespective of the specific privacy guarantee, we point out that for alternative versions of DP (such as zCDP), the acceptance probability results in Proposition 3.1 may no longer hold, as this result depends on the $\epsilon$-DP guarantee. Developing alternatives to Proposition 3.1 for different privacy definitions is another goal of future work.

## Acknowledgments and Disclosure of Funding

R. Gong is supported in part by the National Science Foundation grant DMS-1916002, V. Rao by the National Science Foundation grants RI-1816499 and DMS-1812197, and J. Awan by the National Science Foundation grant SES-2150615.

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
