## Supplemental Materials: Data Augmentation MCMC for Bayesian Inference from Privatized Data

## S-1    Statement on Societal Impacts

We do not foresee direct negative societal impact from the current work. Admittedly, our method is based on imputing the confidential database which privacy mechanisms seek to protect. We can assure the reader that such imputations are based on formally differentially private data products and hence do not violate differential privacy. Also, one may argue that our work is catalytic to enhancing the 'disclosure risk' of individuals, i.e. an adversary might be able to make accurate posterior inference about an individual if the adversary has highly informative and correct prior and modeling information to begin with. Granted, no existing privacy frameworks can guard against this.

## S-2    Proofs in Section 3.1

**Proposition 3.1.** *For a pure $\epsilon$-DP privacy mechanism $\eta$, the acceptance probability $\alpha$ from Equation* (5) *satisfies* $\alpha(x_i^\star \mid x_i, x_{-i}, \theta) \geq \exp(-\epsilon)$, *for all $\theta, x_{-i}, x_i, x_i^*$.*

*Proof.* Step 2a of Algorithm 1 proposes a new state $x_i^\star$ for the i-th record $x_i$ according to the model $f(\cdot \mid \theta)$. Notice that the proposed latent database $x^\star = (x_i^\star, x_{-i})$ and the current latent database $x = (x_i, x_{-i})$ differ in only one entry. Then, the probability of accepting a proposed state $x_i^\star$ is $\alpha(x_i^\star \mid x_i, x_{-i}, \theta) = \min\left(\eta_\epsilon(s_{dp} \mid x^\star)/\eta_\epsilon(s_{dp} \mid x), 1\right)$. This ratio compares two adjacent databases $x^\star$ and $x$. $\epsilon$-DP guarantees that the probability ratio of any output is within $\exp(\pm\epsilon)$ for adjacent databases by Equation (1). □

**Proposition 3.2.** *The Gibbs sampler described in Algorithm 1 requires $O(n)$ number of operations to update the full latent database according to $p(x \mid \theta, s_{\mathrm{dp}})$.*

*Proof.* We prove that each update for $x_i \mid x_{-i}, \theta, s_{\mathrm{dp}}$ is $O(1)$ and hence the full sweep for the latent database $x \mid \theta, s_{\mathrm{dp}}$ is $O(n)$. Given current state $(x, \theta)$, in Step 2a, the method proposes from $x_i^\star \sim f(\cdot \mid \theta)$ independent of other entries $x_{-i}$ and the current state $x_i$; the runtime of this local proposal step does not depend on $n$. Since $\eta(s_{\mathrm{dp}} \mid x)$ is record-additive (Assumption 2), then $t(x^\star, s_{\mathrm{dp}})$ can be computed in $O(1)$ time by $t(x^\star, s_{\mathrm{dp}}) = t(x, s_{\mathrm{dp}}) - t_i(x_i, s_{\mathrm{dp}}) + t_i(x_i^\star, s_{\mathrm{dp}})$ of Step 2b. The density evaluations in Step 2c are also $O(1)$. Overall, to update all $x_i$, $i = 1, 2, \ldots, n$, the runtime is $O(n)$. □

## S-3    Proofs in Section 3.2

### S-3.1    Ergodicity

In Algorithm 2, we first present a Metropolis-within-Gibbs sampler that is more general than Algorithm 1. We prove its ergodicity in Theorem S-3.1, which implies Theorem 3.3.

The Metropolis-within-Gibbs sampler in Algorithm 2 consists of alternating Metropolis-Hastings steps targeting $p(\theta \mid x, s_{\mathrm{dp}}) = p(\theta \mid x)$ and $p(x \mid \theta, s_{\mathrm{dp}})$. In Assumption 1 we have assumed that a Markov kernel for $p(\theta \mid x)$ exists. A typical kernel involves first proposing from some distribution $q_\theta(\theta \mid x)$ and then accepting or rejecting the proposed state an appropriate probability. The data-augmentation steps consist of the sequence of updates $p(x_i \mid x_{-i}, \theta, s_{\mathrm{dp}})$, for $i = 1, 2, \ldots, n$. Algorithm 1 suggests using the proposal $x_i^\star \sim f(\cdot \mid \theta)$ independent of current state $x$. In this more general sampler, described in algorithm 2, we use proposals $q_x(x_i^\star \mid x_i, x_{-i}, \theta, s_{\mathrm{dp}})$ that can depend on current states of $x$ and $\theta$, as well as the private query $s_{\mathrm{dp}}$. Notice that since latent records are exchangeable in both $f(x \mid \theta)$ and $\eta(s_{\mathrm{dp}} \mid x)$, respectively by the i.i.d. model assumption and by record-additivity, it is sufficient to use the same kernel $q_x$ for all $x_i$.

**Theorem S-3.1.** *Under conditions A1 - A4 below, the Gibbs sampler of Algorithm 2 on the joint space $(\mathbb{X}^n \times \mathbb{R}^p)$ is ergodic and it admits $\pi(x, \theta)$ as the unique limiting distribution.*

*A1. The prior distribution is proper and $\pi_0(\theta) > 0$ for all $\theta$ in $\Theta = \{\theta \mid f_\theta(x) > 0$ for some $x\}$.*

**Algorithm 2** A general Metropolis-within-Gibbs sampler for $p(\theta, x \mid s_{\mathrm{dp}})$

---

1. Conditional update of $p(\theta \mid x)$:
   (a) Propose $\theta^\star \sim q_\theta(\theta^\star \mid \theta, x)$.
   (b) Accept $\theta^\star$ with probability

$$\alpha(\theta^\star \mid \theta, x) = \min\left\{\frac{q_\theta(\theta \mid \theta^\star, x)p(\theta^\star)\prod_{i=1}^{n} f(x_i \mid \theta^\star)}{q_\theta(\theta^\star \mid \theta, x)p(\theta)\prod_{i=1}^{n} f(x_i \mid \theta)}, 1\right\}$$

2. For each $i = 1, \ldots, n$, update $p(x_i \mid x_{-i}, \theta, s_{\mathrm{dp}})$ by:
   (a) Propose $x_i' \sim q_x(x_i^\star \mid x_i, x_{-i}, \theta, s_{\mathrm{dp}})$,
   (b) Accept the proposed state $x_i^\star$ with probability

$$\min\left\{\frac{q_x(x_i \mid x_i^\star, x_{-i}, \theta, s_{\mathrm{dp}})\eta(s_{dp} \mid x_i^\star, x_{-i})f(x_i^\star \mid \theta)}{q_x(x_i^\star \mid x_i, x_{-i}, \theta, s_{\mathrm{dp}})\eta(s_{dp} \mid x_i, x_{-i})f(x_i \mid \theta)}, 1\right\}.$$

---

A2. *The model is such that the set $\{x : f(x \mid \theta) > 0\}$ does not depend on $\theta$.*

A3. *The privacy mechanism satisfies $\eta(s_{\mathrm{dp}} \mid x) > 0$ for all $x \in \mathbb{X}^n$.*

A4. *From a valid current state, the proposal kernels satisfies (a) $q_\theta(\theta^\star \mid x, \theta) > 0$ for all $\theta^\star \in \Theta$, and (b) $q_x(x_i^\star \mid x_i, x_{-i}, \theta, s_{\mathrm{dp}}) > 0$ for all $x_i^\star$ with $f(x_i^\star, x_{-i} \mid \theta) > 0$.*

*Proof.* It is sufficient to show that the chain is $\pi$-invariant, aperiodic, and $\pi$-irreducible [Tierney, 1994]. The Metropolis-within-Gibbs sampler is aperiodic by construction, since some proposals can be rejected. It is also $\pi$-invariant because it is composed of kernels that satisfy detailed balance with respect to $\pi$.

Irreducibility means that, informally, every set $A$ with $\pi(A) > 0$ can be reached by the Gibbs sampler from any starting point within finitely many steps. We first prove irreducibility for $n = 1$ and generalize this to a sample size of $n \geq 2$. Suppose $A \subset \mathbb{X}^1 \times \Theta$ with $\pi(A) > 0$ and suppose the current state of the Gibbs chain is $(x^{(0)}, \theta^{(0)})$. For any state $(x, \theta) \in A$ we have $q(\theta \mid x^{(0)}, x^{(0)})q(x \mid x^{(0)}, \theta, s_{\mathrm{dp}}) > 0$ by A4. The acceptance ratios are also positive by A1-A4. As a result

$$P(A \mid x^{(0)}, \theta^{(0)})$$
$$\geq \int\int_A q(\theta \mid x^{(0)}, x^{(0)})q(x \mid x^{(0)}, \theta, s_{\mathrm{dp}})\alpha(\theta \mid x^{(0)}, x^{(0)})\alpha(x \mid x^{(0)}, \theta, s_{\mathrm{dp}})dxd\theta > 0.$$

So when $n = 1$, we can reach $A$ from any starting point in one iteration of the Gibbs sampler. For $n \geq 2$, we can reach the set $A$ in at most $n$ steps: the first iteration moves $x_1$ and $\theta$ into $A$, and subsequent steps moves other $x_i$'s into $A$ while keeping all previous $x_j$'s inside $A$ by rejecting proposals leaving $A$. $\square$

A4 details conditions on the proposal distributions to ensure ergodicity of Algorithm 2. It can be relaxed so long as $\pi$-irreducibility is satisfied. Also, A4a should be viewed as a condition implied by the validity of a kernel targeting $p(\theta \mid x)$ from Assumption 1 and, therefore, is not an additional assumption. Importantly, conditions in A4 are mild because they cover common proposal distributions; Gaussian random walk on $\theta$ for A4a and the independent Metropolis proposals $f(\cdot \mid \theta)$ for A4b are such examples. In Algorithm 1, we use the kernel $q_x(x_i^\star \mid x_i, x_{-i}, \theta, s_{\mathrm{dp}}) = f(x \mid \theta)$, which satisfies $f(x^\star \mid \theta) > 0$ by A2. Hence Theorem S-3.1 implies Theorem 3.3.

## S-3.2 Geometric ergodicity of Algorithm 1

**Theorem 3.4.** *Assume that in step 1 of Algorithm 1, one can directly sample from $p(\theta \mid x)$. Under A1-A3 of Theorem 3.3, the resulting $(x, \theta)$ chain, as well as the marginal chains, are geometrically ergodic if $\eta$ satisfies $\epsilon$-DP and there exists $0 < a \leq b < \infty$ such that $a \leq f(x \mid \theta) \leq b$ $\forall \theta, x$.*

*Proof of Theorem 3.4.* The assumption of $a \leq f(x \mid \theta) \leq b$ leads to the inequality

$$p(\theta \mid x) = \frac{p(\theta)f(x \mid \theta)}{\int p(\theta')f(x \mid \theta')d\theta'} \geq \frac{a}{b}p(\theta),$$

since $p(\theta)$ is a proper prior by A1 of algorithm 1.

This proof proceeds by verifying the drift and minorization conditions of the marginal Markov transition kernel on $X$ according to Theorem 8 of Johnson et al. [2013]. We first present a full proof for $n = 1$ and then generalize the arguments to $n \geq 2$. In this proof, we abbreviate $\eta(s_{\mathrm{dp}} \mid x)$ as $\eta(x)$.

Recall that the probability of accepting proposed state $x^\star$ is $\alpha(x^\star \mid x, \theta) = \min\left(1, \frac{\eta(x^\star)}{\eta(x)}\right)$. The probability of accepting any proposal from the current state is $\alpha(x, \theta) = \int \alpha(x^\star \mid x, \theta)f(x^\star \mid \theta)dx^\star$. Let $K(x' \mid x, \theta)$ denote the Markov transition kernel with respect to the proposal $x^\star \sim f(\cdot \mid \theta)$, and let $K(x' \mid x) = \int K(x' \mid x, \theta)p(\theta \mid x)d\theta$ be the marginal kernel, which integrates out the exact $\theta$ update from $p(\theta \mid x)$. We have

$$K(x' \mid x, \theta) = f(x' \mid \theta)\alpha(x' \mid x, \theta) + (1 - \alpha(x, \theta))\delta_x(x'),$$

where $\delta_x(x')$ is the Dirac-delta function. Then the marginal transition kernel satisfies

$$K(x' \mid x, \theta) \geq f(x' \mid \theta)\alpha(x' \mid x, \theta) \geq a\exp(-\epsilon)$$

since $a(x' \mid x, \theta) \geq \exp(-\epsilon)$ according to Proposition 3.1. As a result, we have

$$p(\theta \mid x)K(x' \mid x, \theta) \geq \frac{a}{b}p(\theta) \cdot a\exp(-\epsilon) \tag{S1}$$

Equation (S1) is sufficient for a minorization condition $K(x' \mid x) \geq a^2 b^{-1} \exp(-\epsilon)$ to hold on $x' \in \mathbb{X}$ since $p(\theta)$ is proper.

To establish a drift condition, let $w : \mathbb{X} \to \mathbb{R}_{>0}$ be integrable with $v = \int w(x)dx < \infty$. Then we have the conditional expectation

$$\begin{aligned}
K_X[w(x)] &= \mathbb{E}\left[w(X^{(t+1)}) \mid X^{(t)} = x\right] \\
&= \int w(x')K(x' \mid x, \theta)dx' \\
&= \int\int w(x')K(x' \mid x, \theta)p(\theta \mid x)d\theta dx' \\
&= \int\int w(x')f(x' \mid \theta)\alpha(x' \mid x, \theta)p(\theta \mid x)d\theta dx' + w(x)\int(1 - \alpha(x, \theta))p(\theta \mid x)d\theta \\
&\leq \int\int w(x')f(x' \mid \theta)p(\theta \mid x)d\theta dx' + w(x)\int p(\theta \mid x)d\theta
\end{aligned}$$

Using $f(x \mid \theta) \leq b$, we can show that

$$K_X[w(x)] \leq bv + w(x), \tag{S2}$$

which is the drift condition. Combining Equations (S1) and (S2), we invoke Theorem 8 of Johnson et al. [2013] to establish geometric ergodicity of the Gibbs sampler.

When $n \geq 2$, the proof shall proceed by denoting $K(x' \mid x, \theta)$ as the Markov transition kernel on $x, x' \in \mathbb{X}^n$ and similarly for $K(x' \mid x)$. The drift condition becomes $K_X[w(x)] \leq b^n v + w(x)$ and minorization condition becomes $K(x' \mid x) \leq (a^2 b^{-1} \exp(-\epsilon))^n$. $\qquad\square$

## S-4  Log-linear Model: More Details

Our full model, along with conjugate priors is given in the following equation array:

$$\text{prior} \qquad\qquad p \sim \text{Dirichlet}(\alpha), \qquad\qquad\qquad\qquad \text{(S3)}$$

$$p_{i-}^k \sim \text{Dirichlet}(\alpha_i^k) \quad \forall i, \qquad\qquad\qquad \text{(S4)}$$

$$\text{data model} \qquad\qquad n_- \sim \text{Multinomial}(N, p_-), \qquad\qquad\qquad \text{(S5)}$$

$$n_{i-}^k \sim \text{Multinomial}(n_i, p_{i-}^k) \quad \forall i, \qquad\qquad \text{(S6)}$$

$$\text{privacy noise} \qquad\qquad L_{ijk} \overset{\text{i.i.d.}}{\sim} \text{Laplace}(0, 2K/\epsilon), \qquad\qquad\qquad \text{(S7)}$$

$$m_{ij}^k = n_{ij}^k + L_{ijk} \quad \forall i,j,k, \qquad\qquad \text{(S8)}$$

$$\text{privatized output} \qquad\qquad s_{dp} = (m_{ij}^k). \qquad\qquad\qquad\qquad\quad \text{(S9)}$$

## S-5  Linear Regression: More Details and Results

**Data generating parameters.** Our experiments use continuous predictors $X_0$, which we model as $X_0^i \overset{\text{i.i.d.}}{\sim} \mathcal{N}_p(m, \Sigma)$. We choose $\Sigma = I_n$. We simulate $m_i \overset{\text{i.i.d.}}{\sim} \mathcal{N}(0,1)$ and hold it fixed at $m = (0.9, -1.17)$.

**Conjugate prior distribution.** Our experiments fix $\sigma^2$ at the data generating value of $\sigma^2 = 2$. Given prior $\beta \sim \mathcal{N}_{p+1}(0, \tau^2 I_{p+1})$, the posterior distribution $\beta \mid \sigma^2, x, y$ is multivariate Normal with covariance matrix $\Sigma_n = (x^\top x/\sigma^2 + I_{p+1}/\tau^2)^{-1}$ and mean vector $\mu_n = \Sigma_n(x^\top y)/\sigma^2$. The prior for $\beta$ is $\beta_i \overset{\text{i.i.d.}}{\sim} \mathcal{N}(0, \tau^2 = 2^2)$.

**The effect of clamping.** We view clamping as part of the privacy mechanism. The clamping step first truncates $x$ and $y$ values into a fixed range, and then performs data-independent location-scale transformation so that all values of $\tilde{x}$ and $\tilde{y}$ are in the range $[-1, 1]$. Although with conjugate priors the confidential data posterior $p(\theta \mid x, y)$ is tractable, the clamped data posterior $p(\theta \mid \tilde{x}, \tilde{y})$ no longer enjoys conjugacy and is now intractable. Since the clamping parameters are known, to sample from the clamped data posterior, one can design data-augmentation MCMC algorithms to impute truncated values. Such an imputation algorithm might take $O(n)$ per iteration. We also highlight that as $\epsilon \to \infty$, in which case privacy noise approaches zero, the posterior $p(\theta \mid s_{\text{dp}})$ approaches $p(\theta \mid \tilde{x}, \tilde{y})$.

**Acceptance rate.** In Section 5, we report the posterior means of $\beta, \beta_1$ and $\beta_2$ given $s_{\text{dp}}$ produced from the same fixed latent database $(x, y)$, with different privacy levels. We also report the acceptance rate of $p(x_i \mid x_{-\theta}, \theta, s_{\text{dp}})$ updates in each iteration of the Gibbs samplers. Recall that for each $s_{\text{dp}}$, we run the Gibbs sampler for 10000 iterations and discard the first half for burn-in. From Figure S1, we can see that the empirical acceptance rate of the IM proposals is much higher than the lower bound of Proposition 3.1.

**Posterior credible intervals.** We repeat the credible interval experiment on log-linear models. First we sample one $\beta$ parameter from the prior, and hold this fixed. Then for each $\epsilon$ value, we produce 100 confidential databases $(x, y)$ and one private $s_{\text{dp}}$ for each non-private one, and then run a chain for 10,000 iterations targeting $\beta \mid s_{\text{dp}}$. After burn-in, from each chain, we produce a 90% credible interval for each $\beta_0, \beta_1$ and $\beta_2$. We then calculate the empirical coverage which is reported in Table 1.

While at $n = 100$, we do not expect the frequentist coverage of the credible intervals to exactly match the nominal level of .9, note that most of the values are close to or above .9. The coverage on $\beta_1$ is lower than 90%, which might be due to the true parameter being furthest from the prior mean of 0. Another explanation is that data quality loss from truncation and location-scale transformations during the clamping procedure can not be fully recovered by our inference procedure.

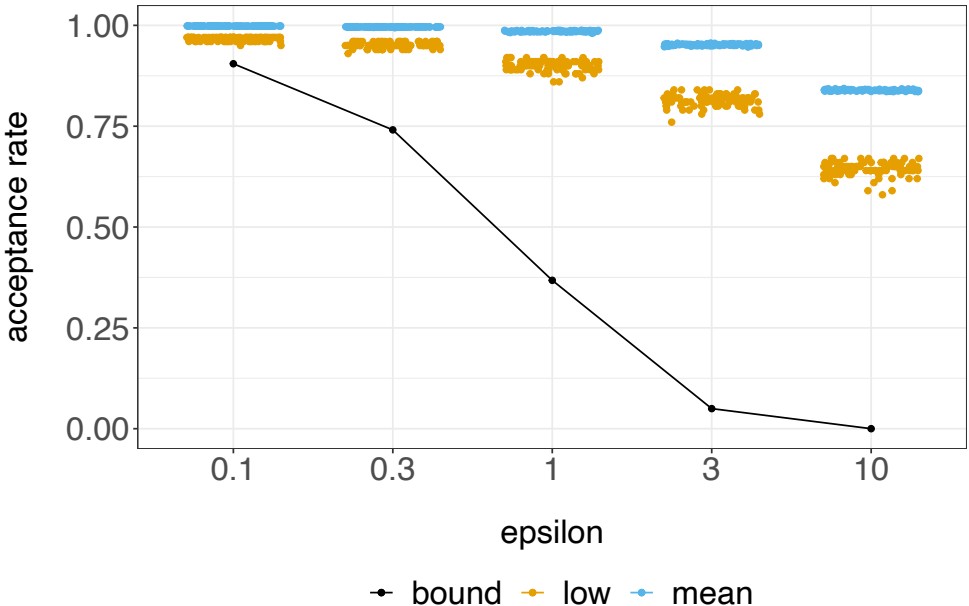

Figure S1: Observed acceptance rates for the log-linear model. The blue (above) point clouds indicate the average acceptance rate, and the orange (below) points indicate the observed minimum acceptance rate of each chain. The solid black line is the lower bound of Proposition 3.1.

| $\epsilon$ | $\beta_0 = -1.79$ | $\beta_1 = -2.89$ | $\beta_2 = -0.66$ |
|---|---|---|---|
| 0.1 | .99 | **.60** | .99 |
| 0.3 | 1 | **.66** | .94 |
| 1 | 1 | **.84** | **.80** |
| 3 | 1 | **.84** | **.75** |
| 10 | .93 | .87 | .85 |

Table 1: Coverage of $\beta_0, \beta_1, \beta_2$ in linear regression. Coverage is based on 100 replicates.

## S-6 Statement on Computing Resources

We ran the experiments on an internal cluster. We used a server with a pair of 64-core AMD Epyc 7662 'Rome' processors and with 256GB of RAM. We ran each MCMC chain for 10000 iterations and a typical chain takes approximately 330 seconds for linear regression and approximately 404 seconds for the log-linear model.