# OpenReview forum: "Data Augmentation MCMC for Bayesian Inference from Privatized Data"
_NeurIPS.cc/2022/Conference — NeurIPS 2022 Accept_

### Official Review · Reviewer_xd2f · 2022-07-07

**Rating:** 7
**Confidence:** 4
**Soundness:** 3 good
**Presentation:** 4 excellent
**Contribution:** 3 good

**Summary:**

In this paper, authors propose a method for noise-aware statistical inference under DP. Essentially, the aim is to enable Bayesian inference that can capture the DP induced noise. Authors propose a novel inference method, that handles the unobserved private data as an additional augmented latent variable to the model. Similar to other recent works on the noise-aware DP Bayesian inference (e.g. Bernstein and Sheldon 2018, 2019 and Kulkarni et al. 2021) the method is based on a single round of data collection from which the posterior of the model parameters is inferred. Authors can bypass the exploding computational cost of introducing tons of latent samples by only considering a particular types of DP data collection methods called (satisfying the Record additivity). By experiments, authors show that the posteriors obtained using the proposed method are well calibrated.

**Questions:**

Some questions:
- It would be interesting to see, how the method compares against the Bernstein and Sheldon '19 solution for the linear regression.
- Also, Kulkarni et al. '21 extend the B&S solution for GLMs by using approximate sufficient statistics. It would be a really interesting to see, would it be possible to write say Logistic regression with the record additivity and see how this compares against the approximate solution of Kulkarni et al.
- What is the number of samples n for the lin. reg. experiment? (I don't think this was mentioned in the manuscript)

Typo: \
Line 228: I think there is an extra "." in the end of the line.



**Limitations:**

In the negatives, I listed a weakness that I think should be better discussed. I think the  potential negative societal impact (or the lack of it) is sufficiently discussed.

**Strengths And Weaknesses:**

Strengths:\
The problem authors are solving is relevant. There have recently been plenty of works aiming to capture the noise introduced by the DP mechanism into the posterior uncertainty of a parameter of a probabilistic model. The main contribution of this work is to solve the problem of latent sensitive data. In previous attempts of noise-aware DP Bayesian inference, the latent data has been presented in some aggregate form such as sufficient stats. in Bernstein and Sheldon '18 and B&S '19, or by approximate sufficient statistics as in Kulkarni et al. '21. Without this assumption, the computational cost of having N (number of samples) latent variables will make the inference infeasible. In this paper, authors have introduced a definition for a general type of privacy mechanism called Record Additivity. DP mechanisms satisfying record additivity, have likelihood in a form that is factorizable over the latent sensitive data. Thus, when authors use Gibbs sampling to propose new samples for the latent variables, they don't need to update the entire log-likelihood expression but can change only the term in the sum that corresponds to the proposed index. This avoids computing the likelihood from scratch every time a new sample is drawn. Authors show that the proposed Metropolis within Gibbs sampler targets asymptotically the correct distribution, and also show the acceptance probability as a function of DP quarantee $\epsilon$. \
$\qquad$ I find this solution very clever, and would be really interested on how wide range of models this approach can actually be extended. In the empirical evaluation, authors consider couple of relatively simple models: a Log-Linear model and a linear regression model. The empirical results mainly demonstrate, that the recovered noise-aware posterior is well calibrated and in terms of mean, follows closely the non-DP solution.

Weaknesses:\
If I am not mistaken, the extent of models that can be solved with this methods is bound to the record additivity. Essentially the functions $t_i$ of Assumption 2, are the way to compare the DP released data to the latent samples. Now, I would imagine $t_i$ should be then characteristic in some sense to actually capture the correct model. I think this should be better discussed.

---

> ### Author Response · Authors · 2022-08-01
> **Response to Reviewer xd2f**
>
> We are very appreciative that Reviewer xd2f saw the merits of our work. We hope that the discussion below will help clarify our contributions.
>
> “If I am not mistaken, the extent of models that can be solved with this methods is bound to the record additivity. Essentially the functions ti of Assumption 2, are the way to compare the DP released data to the latent samples. Now, I would imagine ti should be then characteristic in some sense to actually capture the correct model. I think this should be better discussed.”
>
> The question is whether the privacy mechanism should be tailored in accordance with the model at hand, and whether some models would or would not lead to a mechanism satisfying record additivity. To perform optimal inference, one may indeed need to tailor the privacy mechanism to the data-generating model. In this case, if the data are conditionally i.i.d. given the parameter (which is already assumed in the paper), then the log-likelihood is a record-additive function. As such, exponential mechanisms based on the log-likelihood would satisfy the record-additivity assumption. Similarly, one could add noise to sufficient (or approximately sufficient) statistics if they can be expressed as a sum over data points, which would also be record-additive (e,g., this is satisfied by exponential family distributions).
>
> We also note that in many instances, the analyst is not the same person as the privacy expert. For example, the US Census adds independent noise to a series of counts (which are all record-additive), and it is up to the data analyst to choose a statistical model and perform the correct inference on the noisy counts. In this case, the mechanism may or may not be optimal for the model of interest, yet we are still tasked with performing valid statistical inference on the noisy counts provided. We will include additional discussion on these points in the revised manuscript.
>
> “It would be interesting to see, how the method compares against the Bernstein and Sheldon '19 solution for the linear regression.”
>
> Since the noisy statistics used by Bernstein & Sheldon are record additive, our procedure can be applied to produce exact posterior inference, whereas their method is only asymptotically correct.
>
> “Also, Kulkarni et al. '21 extend the B&S solution for GLMs by using approximate sufficient statistics. It would be a really interesting to see, would it be possible to write say Logistic regression with the record additivity and see how this compares against the approximate solution of Kulkarni et al.”
>
> The approximate sufficient statistics for logistic regression, mentioned in Kulkarni et al 21 are indeed record additive, since they add noise to a statistic, which sums over the data points. So, (similar to the linear regression case above) using the same noisy statistics, we would be able to perform exact posterior inference, whereas the approach of Kulkarni et al is only asymptotically correct.
>
> “What is the number of samples n for the lin. reg. experiment? (I don't think this was mentioned in the manuscript)”
>
> The number of samples n is 100. This is mentioned in the supplementary materials, but we will include this information in the revised paper.

---

### Official Review · Reviewer_5G7A · 2022-07-11

**Rating:** 5
**Confidence:** 4
**Soundness:** 3 good
**Presentation:** 2 fair
**Contribution:** 2 fair

**Summary:**

This paper presents a general MCMC algorithm for approximating the joint posterior distribution of model parameters and true/denoised data, given a corrupted data set that has been privatized using an $\epsilon$-differential privacy mechanism. The algorithm follows the standard approach for Bayesian inference under measurement error; it iterates between re-sampling the true data as a latent variable given parameters, and re-sampling parameters given the true data. The paper proposes a pointwise Metropolis-Hastings step as the transition kernel for re-sampling the true data, and shows that, under some mild conditions on the noise mechanism, the acceptance probability of this step is lower-bounded by $\exp(-\epsilon)$, which is non-vacuous for standard choices (e.g., $\epsilon=1$). The paper also states the $O(n)$ time complexity of this step and presents a derivation of MCMC's ergodicity in this context. Two cases studies, using a Naive Bayes log-Linear model and Bayesian linear regression, substantiate the theoretical claims empirically.

**Questions:**

* Please see the "Strengths and Weaknesses" section for my comments; if I have understated the novelty of any aspect of the paper, please let me know.

* Proposition 3.2 look like it might follow simply from the fact that the M-H algorithm loops over $n$ data points, re-sampling them pointwise---can you provide some intuition why this result is surprising or challenging to obtain?

**Limitations:**

The paper is forthright about the assumptions and limitations of the general approach, in particular the assumed structure of the noise distribution. The paper also addresses the possible societal implications of providing an algorithm that "appears" to de-privatize privatized data; they correctly state that the method cannot "undo" the privacy guarantee, it can only obtain the tightest possible uncertainty about the underlying data, subject to that guarantee.

**Strengths And Weaknesses:**

## Originality

Much of the theory in this paper has been presented before or follows directly from previous literature.

First, this paper seems to miss its connection to the broad category of Bayesian measurement error modeling. The paper presents "data augmentation MCMC" (where the true/denoised data is re-sampled as a latent variable in MCMC) as a novel idea, and dedicates lots of space to its exposition and theory (e.g., proving ergodicity in Theorems 3.3-3.4). However, this general idea and the theory about it is already well-known to the applied Bayesian statistics community, having been introduced by Richardson and Gilks (1993) [1] (or perhaps earlier), and since becoming standard in some probabilistic programming packages (for instance, see the [instructions](https://mc-stan.org/docs/2_21/stan-users-guide/bayesian-measurement-error-model.html) in the Stan User's Guide on how to build a Bayesian measurement error model).

EDIT: The authors have responded to this point. Their setting departs from this setting in that the noise in Bayesian measurement models is usually added pointwise to data points independently (i.e., the "local model"), whereas here the noise can be added as in the "central model".

The translation from general Bayesian measurement error modeling to differential privacy is also not new. The specific case where the measurement error comes from a known $\epsilon$-differential privacy mechanism, is given in section 2 of Schein et al. (2019) [2], who formulate the general problem of "locally private Bayesian inference" and describe solution of data-augmented MCMC; this paper misses that citation.

From what I can tell, the novelty in this paper is restricted to the formulation of an M-H algorithm that is general to different noise mechanisms, and Proposition 3.1 which bounds the acceptance probability. I really appreciate both of these contributions, and encourage the authors to rewrite the paper to center around their practical applications. However, the general conceptual setup, and Theorems 3.4-3.5, which collectively take up the majority of the paper, all follow directly from existing work on Bayesian measurement error modeling.

[1] Richardson, Sylvia, and Walter R. Gilks. "A Bayesian approach to measurement error problems in epidemiology using conditional independence models." American Journal of Epidemiology 138.6 (1993): 430-442.

[2] Schein, Aaron, Zhiwei Steven Wu, Alexandra Schofield, Mingyuan Zhou, and Hanna Wallach. "Locally private bayesian inference for count models." In International Conference on Machine Learning, pp. 5638-5648. PMLR, 2019.

## Quality

I think this paper is completely correct. I read it carefully and checked the proofs. It's only drawback is its missed citations and the fact that many of its proofs and ideas are not new.

## Clarity

The paper is very well-written and clear. The technical exposition is well-organized and precise.

## Significance

I think the proposed algorithm is elegant and generally applicable. One can easily imagine rolling it in to existing probabilistic programming frameworks (e.g., Stan), and providing a modular option for differentially private Bayesian inference. However, the majority of the paper is dedicated to theory and exposition of ideas that are not new; I don't think the paper in its current form will be significant for that reason. I encourage the authors to rewrite this paper, focusing more on the practical/applied impact of this general algorithm, which I believe could have fairly significant impact.

---

> ### Author Response · Authors · 2022-08-01
> **Response to Reviewer 5G7A**
>
> Thank you for your comments and suggestions, which we address below.
>
> “Much of the theory in this paper has been presented before or follows directly from previous literature. First, this paper seems to miss its connection to the broad category of Bayesian measurement error modeling.”
>
> We thank 5G7A for pointing out the connection to the measurement error literature, and we will discuss this connection in the revision. However, there is a clear difference between our setup and measurement error models: in measurement error models (and in local-DP), a noisy data point is usually observed for each of the raw data points, and the observed data points are typically conditionally independent given their corresponding original datapoint; however, in our paper we allow for central model mechanisms which could produce a single low dimensional privatized statistic of the entire dataset, resulting in a complex dependence structure on the latent space. This distinction makes the inference problem significantly more challenging -- while there have been several algorithms to perform Bayesian inference in the local model, there have been relatively few that are applicable in the central model (see Related Literature section). Our algorithm works in this context only because we are able to exploit the privacy guarantee, and the record-additive structure, both of which are novel contributions.
>
> “The specific case where the measurement error comes from a known ϵ-differential privacy mechanism, is given in section 2 of Schein et al. (2019) [2], who formulate the general problem of "locally private Bayesian inference" and describe solution of data-augmented MCMC; this paper misses that citation.”
>
> Thank you for pointing out this reference that we will cite in the revision. We point out that, as discussed above, moving from the local DP model to the central model involves a significantly more challenging sampling problem. As such, the methods of the above paper are not applicable for the general setting considered in our paper. Furthermore, the above paper is focused on a particular privacy mechanism / model, whereas our algorithm is a general wrapper that is applicable for a wide range of privacy mechanisms and models.
>
> “Proposition 3.2 look like it might follow simply from the fact that the M-H algorithm loops over n data points, re-sampling them pointwise---can you provide some intuition why this result is surprising or challenging to obtain?”
>
> Our novel concept of record-additivity is the key component to reducing the runtime to O(n), even in the central model. Without record-additivity, or with a more naive implementation, the  one-at-a-time Gibbs sampler would result in O(n^2) running time in the central model, since updating each data point requires re-evaluating the density of the privacy mechanism (which typically takes O(n) time).
>
> “However, the majority of the paper is dedicated to theory and exposition of ideas that are not new; I don't think the paper in its current form will be significant for that reason. I encourage the authors to rewrite this paper, focusing more on the practical/applied impact of this general algorithm, which I believe could have fairly significant impact.”
>
> “I think the proposed algorithm is elegant and generally applicable. One can easily imagine rolling it into existing probabilistic programming frameworks (e.g., Stan), and providing a modular option for differentially private Bayesian inference.”
>
> In the revision, we will put greater emphasis on the practical contributions of our algorithm, specifically in the ease of use of the sampler as a wrapper to existing MCMC methods. Nevertheless, we still believe that it is important to include the theory on ergodicity and acceptance rate, which support the performance of the Gibbs sampler.
>
> We are thankful that Reviewer 5G7a saw several positive aspects of our work, including its ease of use, wide applicability, and potential impact. We disagree that methods for measurement error models are easily applied to the analysis of privacy-protected data (especially in the central model), and moreover those methods do not exploit the privacy guarantee in the way that our approach does.

---

### Official Review · Reviewer_5yua · 2022-07-11

**Rating:** 3
**Confidence:** 5
**Soundness:** 3 good
**Presentation:** 2 fair
**Contribution:** 2 fair

**Summary:**

This paper introduces an MCMC algorithm for private parameters by augmenting the parameters space with the private date. They propose an MH  and Gibbs sampling type sampler that alternatively updates unobserved variables and parameters. They discuss the acceptance rate and mixing properties for the private setting and provide some simulations to support their claims.

**Questions:**

- Some justifications regarding the contributions and advantages of this method over others are needed.
- How does higher privacy results in a better exploration?
- This algorithm is exact when k->\infty. How would privacy be affected in practice when we cut it off?
how does this method compare in high dimension to fast algorithms such as the one in "On Connecting Stochastic Gradient MCMC and Differential Privacy" by Li et al?
- What is the evaluation metric exactly?


**Limitations:**

- This work suffers from the lack of scalability due to MH step.
- It seems like it is incremental and there is not enough justification why this method should be chosen over others that are proven to be faster.
- Even though the arguments for the proofs are sound but it is nothing novel and it is aligned to what is in the literature.


**Strengths And Weaknesses:**

Strengths:
- It is an interesting problem
- it is easy to follow.

Weakness:
- Experimental results are not enough and do not comprehensively back up their claims. For instance, for a high dimensional setting (both parameter space and unobserved space), how would complexity be affected? Since this is an MCMC algorithm, it seems that as we go to higher dimensions it suffers from a lack of exploring state space. How would privacy be affected? Also, comparison to other MCMC-based privatized methods is required both from a complexity and performance point of view. Bigger datasets such as MNIST can be used as has been by other papers. Comparison to Differentially private MCMC and Faster Differentially Private Samplers via Rényi Divergence Analysis of Discretized Langevin MCMC.
- Contribution seems to be very minor and along with the literature. How does it compare to "Faster Differentially Private Samplers via Rényi Divergence Analysis of Discretized Langevin MCMC"?
- The mixing property and acceptance rate proofs are very standard along with that of MH and other privatized MCMC techniques.
- This paper is highly incremental and seems to be an application of doubly interactable [Murray 2012]. The privacy addition is a very minor contribution and is aligned with the literature.

---

> ### Author Response · Authors · 2022-08-01
> **Response to Reviewer 5yua**
>
> Thank you for you comments. We have addressed them all below.
>
> “...[I]ncremental…not enough justification why this method should be chosen over others that are proven to be faster.”
> “How does it compare to [GK20]? “
> “[C]omparison to other MCMC-based privatized methods is required…”
>
> We believe Reviewer 5yua has confused two very distinct problems related to privacy: using MCMC to achieve privacy, versus using MCMC to analyze privacy-protected data. MCMC-based privatization methods seek to develop samplers to produce privatized statistics. The paper “Faster…” recognizes that some privacy mechanisms (such as the exponential mechanism) are difficult to sample exactly and quantifies the additional privacy cost of using an approximate sampler. The other paper mentioned [Li et al. 2019] shows that the randomness in an MCMC procedure can by itself satisfy DP. In either case, these papers use MCMC as part of a privacy mechanism, which results in a privatized statistic.
>
> Our paper addresses a very different problem: we are given the privatized statistic, and seek to perform statistical inference for forparameters underlying the non-private data. Our paper is not developing a privacy mechanism, but rather is focused on post-processing procedures to analyze privacy-protected data. Due to the very different aims, we do not believe our paper is comparable to any of the listed works above.
>
> Finally, we emphasize that Langevin dynamics has never been applied to draw inference from privacy-protected data, and is not a viable baseline for our task for two reasons: It  1) Langevin dynamics evaluate the derivative of the log-density, which is intractable for our problem, and 2) Langevin dynamics typically do not target the correct posterior distribution. In contrast, our privacy-aware Gibbs sampler is easy to implement and targets the correct posterior distribution.
>
> "...[H]ighly incremental...application of Murray [2012], and that the privacy addition is very minor".
>
> We acknowledge that oursampler uses ideas from the doubly-intractable MCMC literature. That said, the techniques of Murray [2012] have not been applied to the analysis of privacy-protected data. We further point out that our procedure 1) is a straightforward wrapper applicable to any statistical model to which a privacy mechanism has been applied, 2) inherits performance guarantees from the privacy guarantees, and 3) targets the correct posterior distribution while imposing mild restrictions on the models or privacy mechanisms.
>
> “The mixing property and acceptance rate proofs are very standard…”
> “... [P]roofs are sound but it is nothing novel…”
>
> The value in our approach is as an easy-to-implement and effective methodology to achieve valid Bayesian inference on privacy-protected data. The value of our theory is not in the development of novel proof techniques, but in proving that our procedure translates privacy guarantees into efficient mixing. That the proofs are not complex is actually a strength of the paper, as they elucidate the connection between the privacy guarantee of the privatized statistic and the performance guarantee of the Gibbs sampler.
>
> “...[F]or a high dimensional setting…how would complexity be affected? Since this is an MCMC algorithm…it suffers from a lack of exploring state space. How would privacy be affected?”
>
> We agree that in high dimensions, our algorithm has the same limitations as other MCMC algorithms. However, our theory shows how privacy guarantees can ensure efficient mixing of the Gibbs sampler. We also point out that the goal of the paper is to enable posterior inference on privatized data when there already exists a useful MCMC sampler for the posterior based on the original data. Due to this, we do not view the dependence on the parameter dimension as a major limitation.
>
> “How would privacy be affected in practice when we cut off? how does this method compare in high dimension to fast algorithms such as Li et al [2019]?”
>
> We repeat that our algorithm does not seek to produce a privatized statistic (it is only a post-processing of privatized data), so the dimension/mixing of the Markov chain has no effect on privacy.
>
> "[H]ow does privacy result in a better exploration?"
>
> Proposition 3.1 shows that the acceptance probability increases as epsilon decreases, resulting in faster exploration. The intuition is that with higher privacy, each data point has a weaker individual effect on the posterior distribution, enabling easier mixing.
>
> “What is the evaluation metric exactly?”
>
> We are not sure what is meant by this question, but the goals of our algorithm are: 1) target the correct posterior distribution, given a privatized statistic, 2) have the complexity of the sampler be similar to non-private samplers for the same model, 3) have the algorithm be easily implementable and applicable to a wide range of models and mechanisms. We have achieved all of these goals, whereas no prior paper has done so.

---

> > ### Comment · Reviewer_5yua · 2022-08-08
> > **Response to authors**
> >
> > Thank you for taking the time to respond. I have read the comments carefully. This paper seems to have minor contributions and most of my concerns still stand. I thus keep my score.

---

### Meta-Review · Area_Chair_H9SZ · 2022-08-28

**Recommendation:** Accept
**Confidence:** Less certain

**Metareview:**

This paper presents a general MCMC algorithm for approximating the joint posterior distribution of model parameters and private data, given output from a differentially private algorithm. The paper provides new tools for Bayesian inference under privacy constraints, which can be useful for the differential privacy community. There are a few suggestions from the reviewers/discussion. First, even though the authors claimed their results are fully general, they should consider toning it down or at least clarifying upfront their "Record Additivity" assumption, which seems non-trivial. As one of the reviewers remarked, one of the limitations of this approach is that it will not scale well with high-dimensional data. The AC suggests the authors add this limitation in the discussion of the paper.

Other comments:
While not critical, the AC also has a question regarding the following part:
- In line 105: the assumption is that the density of the mechanism's output distribution is known. This has nothing to do with what privacy variant you use to analyze the mechanism.

**Award:**

No

---

### Decision · Program_Chairs · 2022-09-14

Accept